# Interrelations between Gut Microbiota Composition, Nutrient Intake and Diabetes Status in an Adult Japanese Population

**DOI:** 10.3390/jcm11113216

**Published:** 2022-06-05

**Authors:** Ayumi Tamura, Masaya Murabayashi, Yuki Nishiya, Satoru Mizushiri, Kiho Hamaura, Ryoma Ito, Shoma Ono, Akihide Terada, Hiroshi Murakami, Jutaro Tanabe, Miyuki Yanagimachi, Itoyo Tokuda, Kaori Sawada, Kazushige Ihara, Makoto Daimon

**Affiliations:** 1Department of Endocrinology and Metabolism, Hirosaki University Graduate School of Medicine, Hirosaki 036-8562, Japan; h20gm137@hirosaki-u.ac.jp (A.T.); m.murabayashi1@gmail.com (M.M.); y.nishiya.int@hirosaki-u.ac.jp (Y.N.); s.mizushiri@gmail.com (S.M.); hamaura09-10@yahoo.co.jp (K.H.); itoryo1110@gmail.com (R.I.); chocokun0216@gmail.com (S.O.); h21gm152@hirosaki-u.ac.jp (A.T.); hirozj8@gmail.com (H.M.); jutatana@hirosaki-u.ac.jp (J.T.); yanagi@hirosaki-u.ac.jp (M.Y.); 2Department of Oral Healthcare Science, Hirosaki University Graduate School of Medicine, Hirosaki 036-8562, Japan; i-tokuda@hirosaki-u.ac.jp; 3Department of Social Medicine, Hirosaki University Graduate School of Medicine, Hirosaki 036-8562, Japan; iwane@hirosaki-u.ac.jp (K.S.); ihara@hirosaki-u.ac.jp (K.I.)

**Keywords:** type 2 diabetes, gut microbiota, nutrients consumed, cluster analysis, *Balutia*, *Bifidobacterium*

## Abstract

Upon food digestion, the gut microbiota plays a pivotal role in energy metabolism, thus affecting the development of type 2 diabetes (DM). We aimed to examine the influence of the composition of selected nutrients consumed on the association between the gut microbiota and DM. This cross-sectional study of a general population was conducted on 1019 Japanese volunteers. Compared with non-diabetic subjects, diabetic subjects had larger proportions of the genera *Bifidobacterium* and *Streptococcus* but smaller proportions of the genera *Roseburia* and *Blautia* in their gut microbiotas. The genera *Streptococcus* and *Roseburia* were positively correlated with the amounts of energy (*p* = 0.027) and carbohydrate and fiber (*p* = 0.007 and *p* = 0.010, respectively) consumed, respectively. In contrast, the genera *Bifidobacterium* and *Blautia* were not correlated with any of the selected nutrients consumed. Cluster analyses of these four genera revealed that the *Blautia*-dominant cluster was most negatively associated with DM, whereas the *Bifidobacterium*-dominant cluster was positively associated with DM (vs. the *Blautia*-dominant cluster; odds ratio 3.97, 95% confidence interval 1.68–9.35). These results indicate the possible involvement of nutrient factors in the association between the gut microbiota and DM. Furthermore, independent of nutrient factors, having a *Bifidobacterium*-dominant gut microbiota may be a risk factor for DM compared to having a *Blautia*-dominant gut microbiota in a general Japanese population.

## 1. Introduction

Type 2 diabetes (DM), a heterogeneous disorder of glucose metabolism characterized by both insulin resistance and pancreatic β-cell dysfunction, is considered to be a multifactorial disease [1,2]. Among the factors that influence its development, the gut microbiota has recently begun to be studied more closely, as its association with the pathophysiology of most chronic diseases, including DM, has been reported in various studies in humans as well as in animal models [3,4,5]. The gut microbiota is the complex community of microorganisms living in the intestinal tracts. It is essential for various physiologic processes and is considered to be an environmental factor responsible for energy metabolism and, consequently, for metabolic disorders [3,4,5]. Various studies have shown an association between the gut microbiota and DM; however, the details of such reports vary, as the taxa associated with DM differ substantially depending on the studies and/or the geographic areas in which they were conducted [6,7,8,9,10,11,12]. The reasons for such conflicting results may stem from environmental factors, which may affect glucose metabolism together with the gut microbiota and are substantially different depending on the study population. Although the factors responsible for the above such conflicting results have not been well-elucidated yet, the nutrients consumed seem to be among these factors. Namely, since intestinal microorganisms metabolize various diet-derived nutrients into a variety of bioactive compounds, which in turn induce various bioactions, not only the proportion of any specific microorganism, but also the amount of specific nutrients consumed may affect the amount of nutrient-derived bioactive metabolites. For example, short-chain fatty acids (SCFAs), metabolites with various antidiabetic biological functions, are produced from fiber consumed and have been reported to be increased in a diet-dependent fashion [13,14,15,16]. Therefore, the amount of nutrients consumed may have metabolic effects that depend on the proportion of their corresponding intestinal microorganisms. Furthermore, since various intestinal microorganisms may not affect glucose metabolism alone, but jointly, analyses of the independent association of each taxon with DM may not always be reliable.

To examine the association between the gut microbiota and DM precisely in a general Japanese population, we investigated this association considering selected nutrients consumed and interactions between intestinal microorganisms examined using cluster analyses.

## 2. Subjects and Methods

### 2.1. Study Population

Participants were recruited from the Iwaki study, a health promotion study of Japanese people aged over 20 years that aims to prevent lifestyle-related diseases and prolong lifespan. The Iwaki study is conducted annually in the Iwaki area of the city of Hirosaki in Aomori Prefecture, northern Japan [17,18]. The Iwaki study is a population-based study that involves annual comprehensive health examinations. Participants were recruited from residents aged over 20 years living in the Iwaki area (number: approximately 9200) through a public announcement. The Iwaki study itself has no inclusion or exclusion criteria. As shown in Figure 1, of the 1148 individuals who participated in the Iwaki study in 2016, 74 individuals over 77 years of age were excluded to minimize age-related diversity in gut microbiota as previously reported [19], as were 55 individuals with incomplete data. After these exclusions, 1019 individuals (403 men, 616 women) aged 52.4 ± 14.1 years were included in the present study.

This study was approved by the Ethics Committee of the Hirosaki University School of Medicine (No. 2016-028, approved on 27 May 2016), and was conducted in accordance with the recommendations of the Declaration of Helsinki. Written informed consent was obtained from all the participants.

### 2.2. Characteristics Measured

Blood samples were collected in the morning under fasting conditions from a peripheral vein. Fecal samples were collected within 3 days prior to the study by using a commercial tube kit (TechnoSuruga Laboratory Co., Ltd., Shizuoka, Japan), and stored at 4 °C until the DNA was extracted, as previously reported [20,21,22]. The gut microbiota composition was investigated by conducting a next-generation sequencing analysis targeting the V3–V4 region of the prokaryotic 16S rRNA genes, as previously described [23]. The proportion of each genus of the gut microbiota is a composition ratio obtained by dividing the number of read counts of each genus by the total number of read counts. The measurements of the gut microbiota were described previously in detail elsewhere [19]. Of the 395 genera detected in our analysis, 15 major genera with a proportion of ≥0.01, which together accounted for 0.857 of total abundance, were evaluated in the present study. Daily nutritional intake was estimated using the brief self-administered diet history questionnaire (BDHQ), which is a well-annotated structured self-administered questionnaire invented for Japanese adults to estimate their daily intakes of energy, and selected nutrients by assessing dietary habits during the preceding month [20,21,22,23,24,25,26,27,28,29,30]. Although the BDHQ gives values for a large number of nutrients consumed, only the values of macronutrients and fibers were used here to simplify this first analysis in order to maximize statistical power and avoid multiple testing problems. Fiber was used since it is a well-established nutrient associated with DM in relation with the gut microbiota, as described.

The following clinical characteristics were also measured: height, body weight, body mass index (BMI), percent body fat (fat (%)), fasting blood glucose (FBG), fasting serum insulin, glycated hemoglobin (HbA1c), systolic (s) and diastolic (d) blood pressures (BP), serum levels of low-density lipoprotein (LDL)-cholesterol, triglyceride (TG), high-density lipoprotein (HDL)-cholesterol, uric acid (SUA), urea nitrogen, creatinine (Cr), and albumin, and activity levels of aspartate transaminase (AST), alanine transaminase (ALT), and γ-glutamyl transpeptidase (γGTP) activities. The percent body fat was measured by applying the bioelectricity impedance method with a Tanita MC-190 body composition analyzer (Tanita Corp., Tokyo, Japan). HbA1c (%) is expressed as the National Glycohemoglobin Standardization Program value. All laboratory testing was performed in a commercial laboratory (LSI Medience Co., Tokyo, Japan) in accordance with vendor protocols. Insulin secretion was evaluated by performing a homeostasis model assessment of *β*-cell function (HOMA-β), based on fasting blood glucose and insulin levels [31]. Insulin resistance was also assessed based by using homeostasis model assessment (HOMA-R) [31].

DM was defined according to the 2010 Japan Diabetes Society criterion (FBG levels of ≥126 mg/dL) [32]. In subjects whose FBG levels were not measured, DM was defined by an HbA1c concentration of ≥6.5%. Those taking medication for DM were also defined as having DM. Hypertension was defined by a blood pressure of ≥140/90 mmHg or the use of anti-hypertensive therapy. Hyperlipidemia was defined by an LDL cholesterol level of ≥140 mg/dL, a triglyceride level of ≥150 mg/dL, or the use of anti-hyperlipidemic therapy. Alcohol intake status (current drinker or non-drinker) and smoking habits (never, past, or current) were determined using a questionnaire.

### 2.3. Statistical Analysis

Data are presented as means ± SD. The statistical significance of the differences in values between two groups (parametric) and case–control associations between groups (nonparametric) were assessed by using analysis of variance (ANOVA) and the χ2 test, respectively. Differences in gut microbiota composition and selected nutrients consumed between diabetic and nondiabetic subjects were also evaluated with adjustment for possible confounding factors. Correlations between the gut microbiota composition and selected nutrients consumed were evaluated using univariate or multivariate (for independent associations) linear regression analyses. For multivariate analyses, among characteristics related to each other and different between the DM and non-DM groups, one characteristic that showed the lowest *p*-value was selected as an independent variable, i.e., age, gender, BMI, systolic blood pressure, and serum levels of TG, UA, Cr and ALT were selected as possible confounding factors. Based on the four genera found to be independently associated with DM, subjects were grouped by hierarchical clustering into four major clusters, each of which seemed to represent subjects with each corresponding dominant genus, and then were plotted using a principal component (PC) analysis. The association between DM and each cluster was evaluated by multiple logistic regression analyses with adjustment for the multiple factors described above. For statistical analyses, HOMA indices, TG, AST, and ALT were log-transformed (ln) to approximate a normal distribution. For the correlation analyses with HOMA-R and HOMA-β, subjects with FBG levels of >140 mg/dL (n = 50) and <63 (n = 42), respectively, were excluded to evaluate such index precisely. A *p*-value of <0.05 was considered statistically significant. All analyses were performed using JMP pro version 16.0 (SAS Institute Japan Ltd., Tokyo, Japan).

## 3. Results

### 3.1. Clinical Characteristics of the Study Subjects

The clinical characteristics of subjects based on their diabetic status are shown in Table 1. In addition to those related to glucose metabolism, most characteristics measured were significantly different between the DM and non-DM groups, i.e., the diabetic subjects were older (61.45 ± 10.8 vs. 51.4 ± 14.1, *p* < 0.001), more obese (e.g., BMI: 25.2 ± 4.2 vs. 22.7 ± 3.3, *p* < 0.001), hypertensive (71.9% vs. 32.9%, *p* < 0.001), and hyperlipidemic (62.5% vs. 36.8%, *p* < 0.001), and showed modestly deteriorated kidney function compared with the non-diabetic subjects. However, the proportion of lifestyle-related characteristics, such as habitual alcohol drinking and smoking, was not significantly different between these groups.

### 3.2. Differences in the Selected Nutrients Consumed and Proportions of Gut Microbiota Genera between the DM and Non-DM Groups

Differences in selected nutrients consumed between the DM and non-DM groups were then evaluated (Table 2). As shown, no differences between these groups were observed for any selected nutrients consumed. These observations indicate that the diabetic subjects did not eat more than the non-diabetic subjects, at least regarding the amounts of macronutrients and total energy.

We then examined differences in the gut microbiota composition between the DM and non-DM groups (Table 2). Among 15 major genera evaluated in this study, the proportions of *Bifidobacterium, Streptococcus, Roseburia*, and *Blautia* were significantly different between the DM and non-DM groups, even after adjustment with multiple possible confounding factors; the proportions of *Bifidobacterium* and *Streptococcus* were increased and those of *Roseburia* and *Blautia* were decreased in the DM group compared with the non-DM group.

### 3.3. Correlation between the Selected Nutrients Consumed and Gut Microbiota Genera

The correlations between selected nutrients consumed by the subjects and their gut microbiota genera were then examined (Table 3). After adjustment for possible confounding factors (age, gender, BMI, sBP, and serum levels of TG, UA, Cr and ALT), the genus *Streptococcus* was significantly positively correlated with energy consumed (β = 0.075, *p* = 0.027), and the genus *Roseburia* was significantly positively correlated with the proportion of carbohydrate and fiber consumed (β = 0.091, *p* = 0.007, and β = 0.092, *p* = 0.010, respectively). In contrast, the genera *Bifidobacterium* and *Blautia* were not correlated with any selected nutrients consumed.

### 3.4. Assessment of Gut Microbiota Genera as Risk Factors for DM

Logistic regression analyses were performed to assess the major gut microbiota genera as risk factors for DM (Table 4). The genus *Bifidobacterium* was identified as a significant risk factor for DM (odds ratio (OR): 1.68, 95% confidence interval (CI): 1.33–2.13), whereas the genera *Roseburia* and *Blautia* were identified as factors negatively associated with DM (OR: 0.54, 95% CI: 0.30-0.96 and OR: 0.45, 95% CI: 0.24-0.86, respectively) even after adjustment for multiple factors as described above. Furthermore, because the genus *Roseburia* was significantly correlated with the proportion of carbohydrate and fiber consumed, the risk posed by the genus *Roseburia* in the gut microbiota was evaluated with further adjustment for such factors; the negative association remained significant even after this adjustment (OR: 0.53, 95% CI: 0.30-0.95, and OR: 0.54, 95% CI: 0.30-0.96 for adjustments with carbohydrate and fiber, respectively).

### 3.5. A Bifidobacterium-Dominant Gut Microbiota Is a Risk Factor for DM

To further evaluate the association of gut microbiota composition with DM, we grouped the subjects based on the four abovementioned genera found to be independently associated with DM by hierarchical clustering into four major clusters, each of which seems to represent subjects in whom the corresponding genus is dominant (Figure 2). The association between DM and each cluster was evaluated by conducting multiple logistic regression analyses with adjustment for the factors described above (Table 5). A logistic regression analysis that used the *Blautia*-dominant group as the reference, which is the most negatively associated genus with DM as shown in Table 4, revealed that the *Bifidobacterium*-dominant group was the only one significantly positively associated with DM after adjustment with multiple factors as described previously (OR 3.97, 95% CI 1.68–9.35).

## 4. Discussion

In this cross-sectional study of a general Japanese population, we found that, after adjustment with multiple possible confounding factors, the proportion of the genera *Bifidobacterium* and *Streptococcus* in the gut microbiota were positively associated with DM, and the proportions of the genera *Roseburia* and *Blautia* were negatively associated with DM. These findings indicate that the genera *Bifidobacterium* and *Streptococcus* are risk factors for DM, whereas the genera *Roseburia* and *Blautia* are protective factors against DM. Next, we evaluated the influences of selected nutrient factors on these associations because the gut microbiota has been reported to be affected by the nutrients consumed [33,34]. The analyses revealed positive correlations between the genera *Streptococcus* and *Roseburia* in the gut microbiota with energy consumed and the amounts of carbohydrate and fiber consumed, respectively, but no correlation between the proportion of the genera *Bifidobacterium* or *Blautia* with any nutrient factors. Together, these findings may indicate that the DM risk posed by increased energy consumption can be blamed, at least in part, on an increase in the proportion of the genus *Streptococcus* in the gut microbiota and that higher carbohydrate and fiber consumption may decrease the risk of DM by increasing the proportion of the genus *Roseburia* in the gut microbiota. In contrast, the associations between the genera *Bifidobacterium* and *Blautia* and DM were independent of selected nutrients consumed, which suggests that there is no effective diet for changing the proportion of these genera in the gut microbiota to reduce the risk of DM.

As described, intestinal microorganisms metabolize various diet-derived nutrients to bioactive compounds, which appear to induce not only positive but also negative metabolic states. For example, indoles produced from tryptophan induce interleukin-22 production, trimethylamine produced from choline is further converted into trimethylamine-oxide, which acts pro-inflammatory, secondary bile acids converted from primary bile acids act anti-inflammatory, and SCFAs produced from dietary fibers are involved in various beneficial metabolic pathways [13,35,36,37,38]. However, although the biological effects of these metabolites have been widely evaluated, the changes in these metabolites in response to nutrients consumed, particularly in relation with the gut microbiota, remain poorly understood. Namely, SCFAs have been reported to be increased in a diet-dependent fashion [13,14,15,16], but they are produced not only by proposed metabolically beneficial microorganisms but also by potentially pathogenic microorganisms [39,40]. Therefore, the observed associations between DM and the gut microbiota depending on nutrients consumed seem to warrant further analyses with more subjects and/or at the level of species or subspecies.

Because the intestinal microorganisms that compose the gut microbiota seem to affect glucose metabolism jointly rather than independently, we then grouped the subjects according to the four genera observed to be independently associated with DM by cluster analysis, which revealed four clusters each corresponding to a dominant genus, to examine such joint effects. The analyses showed that subjects with a *Bifidobacterium*-dominant gut microbiota were highly associated with having DM, whereas subjects with a *Blautia*-dominant gut microbiota were substantially negatively associated with having DM. Interestingly, as shown in Figure 1, when the subjects were plotted using a PC analysis, subjects with genera *Bifidobacterium-* and *Blautia*-dominant gut microbiota were not plotted fully on opposite sides, which indicates that each genus does not become a risk factor for DM on its own, but rather jointly, thus supporting the importance of analyses with mutual influences among microorganisms in the gut microbiota.

Although the associations of the genera *Streptococcus*, *Roseburia,* and *Blautia* with DM seem to be mostly concordant with the findings of prior studies [3,4,5], the observed positive association between genus *Bifidobacterium* and DM does not agree with most previous reports [6,7,8,9,10,11]. This conflicting result may stem from differences in the methodology used and/or study population. Notably, the proportion of genus *Bifidobacterium* in the gut microbiota appears to be substantially different depending on the study population. Study populations that showed a negative association between genus *Bifidobacterium* and DM generally had a relatively low proportion of genus *Bifidobacterium* (0.019~0.072) [6,7,8], while one unique study with opposite results used a Japanese population with a relatively high proportion of genus *Bifidobacterium* in the gut microbiota (~0.16) [12]. These findings suggest the possibility of a U-shaped association between genus *Bifidobacterium* and DM, i.e., a negative association when the proportion of genus *Bifidobacterium* in the gut microbiotas of the study population is low, but a positive association when the proportion of genus *Bifidobacterium* in the gut microbiotas of the study population is high. Our study population belongs to the same ethnicity as the study population described above that showed a positive association between genus *Bifidobacterium* and DM and had a similarly relatively high proportion of genus *Bifidobacterium*, 0.080. Furthermore, correlation analyses of genus *Bifidobacterium* with indices representing glucose metabolism, such as FBG and HOMA-R, indicated such U-shaped associations with inflection points at 0.149 and 0.133, respectively (Figure 3). Together, an increased proportion of genus *Bifidobacterium* in the gut microbiota seems to be a risk factor for DM, at least in populations with a high proportion of this genus, or Japanese populations.

The effect of dietary intervention on glucose metabolism is often explained as being partly caused by changes in the gut microbiota, although the reported results are conflicting [7,11,12,41,42]. A recent systematic review assessing such effects in subjects with DM reported no significant changes in the proportions of bacteria in genera including *Bifidobacterium* and *Roseburia* upon the dietary intervention [43]. Similarly in our study, the proportion of genus *Bifidobacterium* in the gut microbiota did not correlate with any nutrient factors, which may suggest that improvement in glucose metabolism by a dietary intervention cannot be simply explained as its effects on increasing the genus *Bifidobacterium* in the gut microbiota. In contrast, the genus *Roseburia* was found to be positively correlated in our study with the proportion of carbohydrate and fiber consumed. *Roseburia* produce short-chain fatty acids (SCFAs) from non-digestible carbohydrates, and SCFAs activate G-protein-coupled receptors (GPCRs), through which SCFAs exert various biological functions that are beneficial for the development of DM such as increasing the levels of glucagon like peptide-1 (GLP-1) and peptide tyrosine-tyrosine (PYY) [44,45]. Therefore, this correlation indicates that dietary intervention to increase the carbohydrate or fiber consumed may improve glucose metabolism at least, in part, by increasing the proportion of the genus *Roseburia* in gut microbiota and, thus, increasing the amounts of SCFAs.

Like the genus *Roseburia*, the genus *Blautia* is also known to produce SCFAs and to be inversely associated with various diseases including DM [19,46,47]. However, no correlations between genus *Blautia* and any selected nutrients consumed were observed in this study. Thus, although an increased proportion of the genus *Blautia* in the gut microbiota exerts substantial effects on the pathophysiology leading to DM, selected nutrients consumed may not have a substantial influence on the proportion of the genus *Blautia* in the gut microbiota, i.e., the genus *Blautia* seems to be a protective factor against DM independent of the amounts of selected nutrients consumed.

The present study has both strengths and limitations. Its strengths include the statistical adjustments made for multiple factors that could have confounded the results and its analysis of a general population-based sample. It also had the following limitations. The participants were selected from a health promotion study, not from a population undergoing ordinary health check-ups, and thus may not accurately represent the general population. Additionally, we examined associations between DM and microorganisms in the gut at the genus level because such associations at the phylum level have been extensively reported. Extended analyses at other levels of taxonomy, such as the species level, may provide even more detailed information on this issue. Furthermore, the influence of the macronutrients consumed on the associations with DM was evaluated to simplify the analyses and avoid the issue of multiple testing. However, the micronutrients consumed may also have some influence on the gut microbiota and its association with DM, and thus analyses of micronutrients consumed should be examined in future work. This study was observational, i.e., no nutrient factor-based interventions were conducted. Therefore, the effects of an intervention using any nutrient factors could not be evaluated precisely. Drugs used by the study participants were also not evaluated, even though some drugs used to treat DM, such as metformin, have been repeatedly reported to alter gut microbiota [11,48]. As the subjects of this study were participants of a health care check-up, more than half of the diabetic subjects (56 out of 96) did not take any drugs to treat DM, and the prevalence of metformin usage has been reported to be much lower in Japan compared with that in other countries such as Taiwan, Hong Kong, and the United States [49] Therefore, the influence of drugs taken by the study participants on the results reported here may not be so substantial. Finally, because our study was cross-sectional and not a cohort study, we could not assess the cause–consequence relationship between the gut microbiota and DM; therefore, cohort analyses are warranted to reveal the relationship between the gut microbiota composition and the incidence of DM.

## 5. Conclusions

The proportion of the genera *Streptococcus* and *Bifidobacterium* in the gut microbiota was positively associated with DM, and those of the genera *Roseburia* and *Blautia* were negatively associated with DM. The observed positive correlation between the proportions of the genera *Streptococcus* and *Roseburia* with energy consumed, and amounts of carbohydrate and fiber consumed, respectively, indicate the possible involvements of nutrient factors in the association between the gut microbiota and DM. Furthermore, independent of nutrient factors, subjects in a general Japanese population with a *Bifidobacterium*-dominant gut microbiota seem to be at higher risk for DM, while those with a *Blautia*-dominant gut microbiota seem to be more protected against DM. The results reported here seem to reinforce the importance of considering nutritional factors in examining the association between gut microbiota and DM and the joint effects of gut microbiota on that association, and may lead to the discovery of which nutritional interventions are appropriate for which individuals in the future.

## Figures and Tables

**Figure 1 jcm-11-03216-f001:**
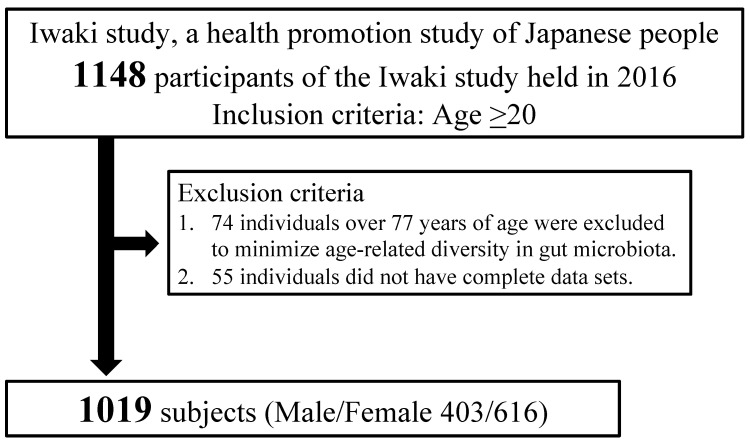
Flowchart of the selection of study participants.

**Figure 2 jcm-11-03216-f002:**
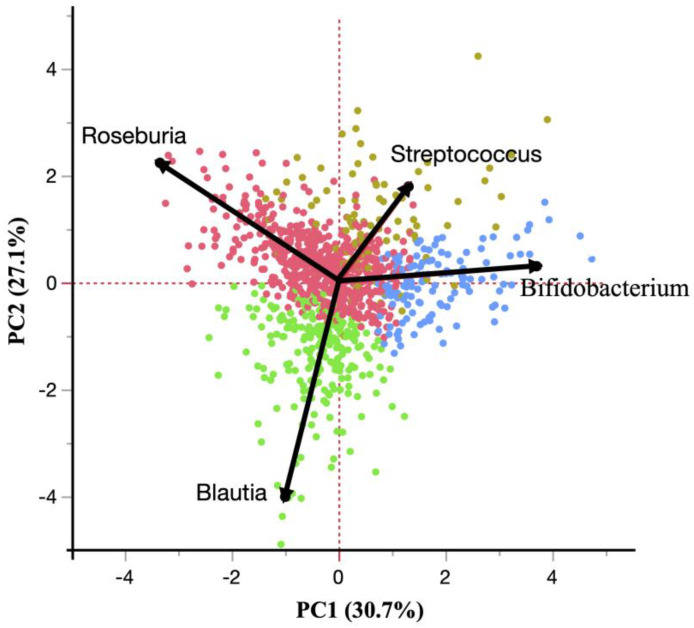
Plots of study subjects clustered by the genera *Bifidobacterium*, *Streptococcus*, *Roseburia,* and *Blautia* using a principal component (PC) analysis. Subjects are plotted in four different colors corresponding to each of four evaluated clusters. Eigenvectors are also shown as arrows with their representing genus names attached.

**Figure 3 jcm-11-03216-f003:**
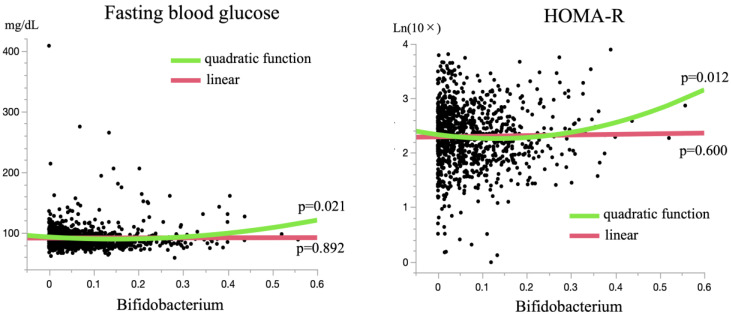
Correlation of genus *Bifidobacterium* with fasting blood glucose and HOMA-R. Regression lines and curves evaluated by linear and quadratic function regression analyses, respectively, are shown with their *p*-values for correlation. Quadratic function regression analyses showed significant correlations, but linear regression analyses did not.

**Table 1 jcm-11-03216-t001:** Clinical characteristics of the study subjects based on diabetic status.

Characteristics	DM (n = 96)	Non-DM (n = 923)	*p*
Number (Gender: M/F)	44/52	359/564	0.188
Age (years)	61.47 ± 10.76	51.43 ± 14.12	<0.001 **
Height (cm)	160.30 ± 9.06	161.35 ± 8.90	0.270
Body weight (kg)	64.72 ± 12.36	59.34 ± 11.27	<0.001 **
Body mass index (kg/m^2^)	25.15 ± 4.15	22.70 ± 3.26	<0.001 **
Percent body fat	29.25 ± 9.70	25.96 ± 7.96	<0.001 **
Fasting plasma glucose (mg/dL)	129 ± 44.43	87.85 ± 9.56	<0.001 **
HbA1c (%)	7.21 ± 1.31	5.71 ± 0.30	<0.001 **
Fasting serum insulin: IRI (mU/mL)	7.30 ± 4.26	5.06 ± 2.62	<0.001 **
HOMA-R	2.31 ± 1.53	1.23 ± 0.63	<0.001 **
HOMA-β	63.75 ± 166.54	81.00 ± 2.13	<0.001 **
Systolic blood pressure (mmHg)	133.5 ± 18.45	123.10 ± 17.45	<0.001 **
Diastolic blood pressure (mmHg)	79.25 ± 10.60	75.09 ± 12.07	0.001 **
LDL cholesterol (mg/dL)	123.41 ± 37.18	117.21 ± 29.15	0.054
Triglyceride (mg/dL)	128.98 ± 105.11	94.98 ± 61.62	<0.001 **
HDL cholesterol (mg/dL)	58.13 ± 17.36	65.52 ± 17.08	<0.001 **
Serum albumin (g/dL)	4.45 ± 0.34	4.51 ± 0.29	0.056
Serum uric acid (mg/dL)	5.44 ± 1.37	5.05 ± 1.35	0.007 **
Serum urea nitrogen (mg/dL)	15.94 ± 6.79	14.00 ± 4.00	<0.001 **
Serum creatinine (mg/dL)	0.84 ± 0.97	0.71 ± 0.18	<0.001 **
AST	26.41 ± 11.00	22.86 ± 9.12	<0.001 **
ALT	29.06 ± 19.42	21.44 ± 13.68	<0.001 **
γGTP	39.69 ± 27.80	33.33 ± 42.63	0.153
Hypertension: n (%)	69 (71.9)	304 (32.9)	<0.001 **
Hyperlipidemia: n (%)	60 (62.5)	340 (36.8)	<0.001 **
Drinking alcohol: n (%)	42 (43.8)	452 (49.0)	0.330
Smoking (never/ past/ current): n	59/18/19	571/183/168	0.917

*p* < <0.01 is indicated by **. Data are presented as the mean ± SD or number of subjects (%).

**Table 2 jcm-11-03216-t002:** DM-dependent differences in nutrients consumed and gut microbiota composition.

			*p* (Adjusted)		
	DM	Non-DM	Non	Age and Gender	Multiple Factors
Energy (kcal/kg/day)	31.755 ± 10.93	31.998 ± 10.23	0.826	0.142	0.444
Carbohydrate (g/kg/day)	4.197 ± 1.45	4.277 ± 1.43	0.599	0.095	0.720
Protein (g/kg/day)	1.123 ± 0.48	1.198 ± 0.50	0.571	0.292	0.440
Fat (g/kg/day)	0.903 ± 0.35	0.907 ± 0.37	0.922	0.567	0.167
Fiber (g/kg/day)					
Total	0.194 ± 0.08	0.189 ± 0.09	0.601	0.107	0.882
Water soluble	0.048 ± 0.02	0.047 ± 0.02	0.783	0.100	0.954
Water insoluble	0.139 ± 0.05	0.136 ± 0.06	0.587	0.108	0.872
Bifidobacterium	0.098 ± 0.11	0.078 ± 0.08	0.024 *	<0.001 *	<0.001 *
Collinsella	0.044 ± 0.05	0.040 ± 0.05	0.435	0.1648	0.339
Bacteroides	0.088 ± 0.10	0.111 ± 0.08	0.010 *	0.048*	0.057
Prevotella	0.060 ± 0.12	0.049 ± 0.10	0.315	0.8522	0.794
Alistipes	0.018 ± 0.03	0.016 ± 0.02	0.359	0.4956	0.413
Gemmiger	0.025 ± 0.02	0.025 ± 0.03	0.858	0.8266	0.884
Streptococcus	0.037 ± 0.06	0.019 ± 0.03	<0.001 **	0.001 **	0.005 **
Roseburia	0.038 ± 0.04	0.046 ± 0.05	0.131	0.022 *	0.020 *
Anaerostipes	0.042 ± 0.05	0.058 ± 0.06	0.017 *	0.1357	0.096
Fusicatenibacter	0.017 ± 0.02	0.021 ± 0.02	0.096	0.1891	0.202
Blautia	0.057 ± 0.04	0.075 ± 0.04	<0.001 **	0.012 *	0.016 *
Ruminococcus ^2^	0.045 ± 0.04	0.051 ± 0.06	0.341	0.6327	0.392
Ruminococcus ^1^	0.034 ± 0.04	0.032 ± 0.05	0.684	0.7635	0.961
Faecalibacterium	0.072 ± 0.05	0.079 ± 0.06	0.231	0.1791	0.315
Lachnospiracea_incertae_sedis	0.019 ± 0.01	0.020 ± 0.01	0.707	0.4297	0.586

*p* < 0.05 and <0.01 are indicated by * and **, respectively. Data are presented as the mean ± SD or number of subjects (%). Multiple factors: age, gender, BMI, sBP, and levels of TG, SUA, Cr, and ALT. Ruminococcus species belong to two different families, Ruminococcaceae and Lachnospiraceae, and, thus, were classified into two corresponding genera, Ruminococcus 1 and 2.

**Table 3 jcm-11-03216-t003:** Correlation between nutrients consumed and gut microbiota genera.

	Bifidobacterium		Streptococcus		Roseburia		Blautia	
	β	*p*	β	*p*	β	*p*	β	*p*
Energy (kcal/kg/day)	−0.017	0.627	0.075	0.027 *	0.043	0.213	−0.019	0.583
Carbohydrate (g/kg/day)	0.004	0.902	0.062	0.063	0.091	0.007 **	−0.022	0.522
Protein (g/kg/day)	−0.023	0.493	0.047	0.164	0.010	0.765	−0.032	0.353
Fat (g/kg/day)	0.015	0.653	0.041	0.215	−0.008	0.822	−0.020	0.557
Fiber (g/kg/day)								
Total	−0.028	0.420	0.015	0.674	0.092	0.010 *	−0.049	0.167
Water soluble	−0.024	0.486	0.004	0.900	0.091	0.010 *	−0.035	0.315
Water insoluble	−0.028	0.453	0.020	0.579	0.069	0.010 *	−0.049	0.171

*p* < 0.05 and <0.01 are indicated by * and **, respectively. Data are presented as the mean ± SD or number of subjects (%). Multiple factors: age, gender, BMI, sBP, and levels of TG, SUA, Cr, and ALT.

**Table 4 jcm-11-03216-t004:** Risk of gut microbiota genera for DM.

	Univariate			Multiple Factors Adjusted		
	OR	95%CI	*p*	OR	95%CI	*p*
Bifidobacterium (per 0.1)	1.28	1.03–1.59	0.026 *	1.68	1.33–2.13	<0.001 *
Streptococcus (per 0.1)	2.27	1.51–3.40	<0.001 **	1.47	0.93–2.32	0.107
Roseburia (per 0.1)	0.68	0.41–1.12	0.132	0.54	0.30–0.96	0.027 *
Blautia (per 0.1)	0.31	0.17–0.58	<0.001 **	0.45	0.24–0.86	0.011*

*p* < 0.05 and <0.01 are indicated by * and **, respectively. Data are presented as the mean ± SD or number of subjects (%). Multiple factors: age, gender, BMI, sBP, and levels of TG, SUA, Cr, and ALT.

**Table 5 jcm-11-03216-t005:** Risk of gut microbiota clustered based on four associated genera for DM.

	Univariate			Age and Gender Adjusted			Multiple Factors Adjusted		
	OR	95%CI	*p*	OR	95%CI	*p*	OR	95%CI	*p*
Blautia dominant	Ref	-	-	Ref	-	-	Ref	-	-
Roseburia dominant	1.72	0.30–1.11	0.101	1.44	0.74–2.80	0.278	1.54	0.78–3.04	0.218
Streptococcus dominant	4.01	1.85–8.70	<0.001 **	2.28	1.02–5.10	0.044 *	2.10	0.90–4.87	0.084
Bifidobacterium dominant	2.57	1.18–5.62	0.018 **	3.43	1.52–7.75	0.003 **	3.97	1.68–9.35	0.002 **

*p* < 0.05 and <0.01 are indicated by * and **, respectively. Data are presented as the mean ± SD or number of subjects (%). Multiple factors: age, gender, BMI, sBP, and levels of TG, SUA, Cr, and ALT.

## Data Availability

All data generated or analyzed during this study are included in this published article.

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
