# Peer review of "Interrelations between Gut Microbiota Composition, Nutrient Intake and Diabetes Status in an Adult Japanese Population"

_jcm, 2022, doi:10.3390/jcm11113216_

Round 1

Reviewer 1 Report

The authors assess the influence of nutrients consumed on the association between the gut microbiota and DM in 1019 Japanese people (403 men, 616 women) aged 52.4 ± 14.1 years by using 16S rRNA sequencing. And in this cross-sectional study, they observed that the proportion of the genera Bifidobacterium and Streptococcus in the gut microbiota were positively associated with DM, and the proportions of the genera Roseburia and Blautia were negatively associated with DM.

Major comments:

  1. The conclusion of this manuscript is not persuasive and convincing. For 16S sequencing can only detect genus level gut microbiota, and although belonging to the same genus, different species may play completely opposite physiological functions, the research on diabetes has gone deep into species level or even subspecies level.
  2. There are numerous studies on diabetes and gut microbiota, as a cross-sectional study with no intervention, this manuscript lacks novelty.

Minor comments:

  1. The manuscript uses tables to present main results, which is slightly simple and not readable for omics articles.
  2. The manuscript uses the brief self-administered diet history questionnaire (BDHQ) to estimate daily nutritional intake, but the questionnaire is not widely used internationally, it may not be easy to ensure the reliability and validity.
  3. As mentioned in discussion, ‘the observed positive association between genus Bifidobacterium and DM does not agree with most previous reports’, this should not put the blame on methodology differences.

Author Response

Dear Reviewer 1

   Thank you very much for having reviewed our manuscript entitled "Influence of nutrients consumed on associations between the gut microbiota composition and type 2 diabetes in a Japanese population: jcm-1717551”. We have changed our manuscript to fulfill your criticisms as much as possible, and in most cases, I just tried to change as you requested.But, in some cases where other reviewers recommended other ways, you may see several changes that you have not suggested. In any case, we beg your generosity to kindly feel satisfied with my responses. The details of the responses to each criticism were written below.

The authors assess the influence of nutrients consumed on the association between the gut microbiota and DM in 1019 Japanese people (403 men, 616 women) aged 52.4 ± 14.1 years by using 16S rRNA sequencing. And in this cross-sectional study, they observed that the proportion of the genera Bifidobacterium and Streptococcus in the gut microbiota were positively associated with DM, and the proportions of the genera Roseburia and Blautia were negatively associated with DM.

----- Thanks for your evaluation. In addition to those you described above, we would like to address the possible involvements of nutrient factors in the association between the gut microbiota and DM, and the importance of considering joint effects of intestinal microorganisms on such association. Please, evaluate also these results, which were the most points we would like to address here.

Major comments:

  1. The conclusion of this manuscript is not persuasive and convincing. For 16S sequencing can only detect genus level gut microbiota, and although belonging to the same genus, different species may play completely opposite physiological functions, the research on diabetes has gone deep into species level or even subspecies level.

-----Yes, surely your comment is appropriate. As you mentioned, we agree extended analyses at other levels of taxonomy, such as the species level, may provide even more detailed information on this issue. However, we were afraid that such analyses were not suitable for the study, since the number of microorganisms appears to increase substantially when extending analyses at the species levels. Namely, this study was conducted as an initial step of this kind of association study and the number of subjects was not substantial, we did not like to analyze too many microorganisms in the gut at once to avoid an issue of multiple testing, or to keep statistical power proper. Besides, we described that the analyses to examine associations between DM and microorganisms in the gut at the genus level as a limitation of the study in the discussion. So, please understand the situation and be generous to accept that what we have done here was, if not best, but at least appreciable at this situation.

  1. There are numerous studies on diabetes and gut microbiota, as a cross-sectional study with no intervention, this manuscript lacks novelty.

-----I believe that our manuscript has, at least, some novelty, as described previously. Namely, what we would like to address the most were the possible involvements of nutrient factors in the association between the gut microbiota and DM, and the importance of consideration of joint effects of intestinal microorganisms on such association. These kinds of analyses have not been conducted so frequently, and, thus, have, at least, some novelty. So, please generously be satisfied to accept that the results reported here deserve to be addressed.

-----For those major comments.

   Anyway, to reply to your comments as much as possible, or to explain more what we would like to address the most, we added the following paragraph and sentences in discussion and conclusion:

   In discussion, 2nd paragraph

As described, intestinal microorganisms metabolize various diet-derived nutrients to bioactive compounds, which appear to induce not only positive but also negative effects on metabolic conditions. For example, indoles produced from tryptophan induce interleukin-22 production, and, trimethylamine produced from choline is further converted into trimethylamine-oxide, which acts pro-inflammatory, while secondary bile acids converted from primary bile acids act anti-inflammatory, and SCFAs produced from fibers are involved in various metabolic pathways beneficially [13, 35-38]. However, although biological actions of these various metabolites have been evaluated extensively, nutrients consumption dependent changes in these metabolites have not well elucidated yet especially in relation with the gut microbiota. Namely, SCFAs have been reported to be increased in a diet-dependent fashion [13-16], but they are produced by not only proposed metabolically beneficial microorganisms but also by potentially pathogenic microorganisms [39, 40]. Therefore, nutrients consumption dependent association between DM and the gut microbiota observed here seem to warrant further analyses with more subjects and/or at the species or even subspecies levels.

   In conclusion at the end

   The results reported here seem to reinforce the importance of considering nutritional factors in examining the association between gut microbiota and DM and the joint effects of gut microbiota on that association, and may lead to discover which nutritional interventions are appropriate for which individuals in the future.

Minor comments:

  1. The manuscript uses tables to present main results, which is slightly simple and not readable for omics articles.

---- Please forgive me if we missed the points of your comment, because we do not speculate how to modify the tables clearly upon your comments. Surely, vertical rules in Table 1 are out of the regulation, and, thus, we deleted them, but what els?. If what we have done are not correct, please, generously specify what we should do. Thanks in advance.

  1. The manuscript uses the brief self-administered diet history questionnaire (BDHQ) to estimate daily nutritional intake, but the questionnaire is not widely used internationally, it may not be easy to ensure the reliability and validity.

----We believe that the BDHQ is well validated and, if not widely, but frequently used in various studies: e.g. Nutrients. 2021 Jul 9;13(7):2345; BMJ Open Respir Res. 2021 Aug;8(1):e000807; Nutrients. 2019 Oct 21;11(10):2540. Anyway, upon your comment, we added these 3 references to explain the methods in the text.

  1. As mentioned in discussion, ‘the observed positive association between genus Bifidobacterium and DM does not agree with most previous reports’, this should not put the blame on methodology differences.

----Please forgive me if I missed the point of your comment. We think that such difference may stems from the difference in population or races, in which proportion of Bifidobacterium in the gut appear to be much abundant compared to those used in others studies. Further, we did not conclude such explanation, as we know that such explanation is just a speculation. Anyway, we believe that such explanation can be deserved to be, at least, mentioned. So, please forgive us for making such comments in discussion.

Reviewer 2 Report

The manuscript presents an interesting cross-sectional study on the association between the gut microbiota and type 2 diabetes where the association in consideration was studied with a selection of macronutrients consumed and with interactions between intestinal microorganisms, examined using cluster analyses.

The manuscript’s topic is interesting and the study is well constructed, however, a few corrections have to be made regarding the explanatory process of results, and some points need to be clarified and corrected.

Title

Please correct the title and add ‘selected’ to describe nutrients examined - „Influence of consumption of selected nutrients on associations between the gut microbiota composition and type 2 diabetes in a Japanese population”. Please correct the phrasing through the body of the manuscript to highlight ‘selected’ nutrients or ‘a selection of’ nutrients.

Introduction

The introduction is quite short and would benefit from more supporting information. Please expand the introduction of the potential impact of the nutrients on gut microbiota and type 2 diabetes and include more references to literature.

Subjects and methods

In the section Study population: please add a new Figure - a flow chart of the selection of study participants.  Please present the criteria for including in/excluding from the study.

In the description of methodology please introduce in detail which nutrients were included in the study and why.

Results

Please also explain why alcohol consumption and smoking were not taken into account as confounding factors. This refers to the results presented in tables 2-5.

Discussion

Please use more references to literature to expand the discussion of the impact of nutrients which when consumed have a positive effect on gut microbiota. Please also discuss the consumption of which nutrients has a negative influence on gut microbiota. In the context of both the positive and the negative impact please highlight the possible mechanisms which explain the results.

Conclusion

At the end, please present the practical aspects of the study.

After implementing the above corrections please verify the abstract as well as make corrections to the relevant sections.

Author Response

Dear Reviewer 2

   Thank you very much for having reviewed our manuscript entitled "Influence of nutrients consumed on associations between the gut microbiota composition and type 2 diabetes in a Japanese population: jcm-1717551”. We have changed our manuscript to fulfill your criticisms as much as possible, and in most cases, I just changed as you requested. But, in some cases where another reviewer recommended another way, you may see several changes that you have not suggested or no change that you have suggested. In any case, we beg your generosity to kindly feel satisfied with my responses. The details of the responses to each criticism were written below.

The manuscript presents an interesting cross-sectional study on the association between the gut microbiota and type 2 diabetes where the association in consideration was studied with a selection of macronutrients consumed and with interactions between intestinal microorganisms, examined using cluster analyses.

The manuscript’s topic is interesting and the study is well constructed, however, a few corrections have to be made regarding the explanatory process of results, and some points need to be clarified and corrected.

------ Thanks, we appreciate your evaluation a lot.

Title

Please correct the title and add ‘selected’ to describe nutrients examined - „Influence of consumption of selected nutrients on associations between the gut microbiota composition and type 2 diabetes in a Japanese population”. Please correct the phrasing through the body of the manuscript to highlight ‘selected’ nutrients or ‘a selection of’ nutrients.

------Thanks. We added the word “selected” according to your suggestion throughout the text including the title.

Introduction

The introduction is quite short and would benefit from more supporting information. Please expand the introduction of the potential impact of the nutrients on gut microbiota and type 2 diabetes and include more references to literature.

----We added several phrases and references here as follows:

Namely, since intestinal microorganisms metabolize various diet-derived nutrients to various bioactive compounds, which in turn induces various bioactions, not only proportion of any specific microorganism, but also amount of specific nutrients consumed may affects to the amount of nutrient-derived bioactive metabolite. For example, short-chain fatty acids (SCFAs), metabolites with various anti-diabetic biological functions, are produced from fibers consumed, and have been reported to be increased in a diet-dependent fashion [13-16]. Therefore, amount of nutrients consumed may exert metabolic effects dependent on proportion of their corresponding intestinal microorganisms.

Subjects and methods

In the section Study population: please add a new Figure - a flow chart of the selection of study participants.  Please present the criteria for including in/excluding from the study.

---We added a flow chart of the selection of study participants together with in/excluding criteria as Figure 1 and, thus, the figures after it were re-numbered accordingly.

In the description of methodology please introduce in detail which nutrients were included in the study and why.

-----Thanks. We added the following sentences in the text:

Although BDHQ give values for huge number of nutrients consumed, we here used values of macronutrients and fibers only to simply this first analysis, to maximize statistical power, and to avoid an issue of multiple testing. Fibers were used, since, as described, they were well-established nutrients related to DM in relation with the gut microbiota.

Results

Please also explain why alcohol consumption and smoking were not taken into account as confounding factors. This refers to the results presented in tables 2-5.

----Thanks for your comments. Alcohol consumption and smoking may have some influence on the association examined. However, as shown in Table 1. Such factors are not significantly different between DM and non-DM subjects, and, thus, we believe that such factors do not have substantial impact on association examined. Anyway, upon your comments, we tentatively analyze some data with alcohol consumption and smoking included as confounding factors, which did not change results substantially as expected: e.g. Table 2, streptococcus, p=0.005→0.005 ï¼›Bacteroides, p=0.057→0.054.Besides, this fact has been described in the methods as “For multivariate analyses, among characteristics related each other and different between the DM and non-DM groups,……” . So, please be satisfied with not taking alcohol consumption and smoking into account as confounding factors.

Discussion

Please use more references to literature to expand the discussion of the impact of nutrients which when consumed have a positive effect on gut microbiota. Please also discuss the consumption of which nutrients has a negative influence on gut microbiota. In the context of both the positive and the negative impact please highlight the possible mechanisms which explain the results.

---Thanks. We added several more references and the sentences to expand the discussion as follows:

As described, intestinal microorganisms metabolize various diet-derived nutrients to bioactive compounds, which appear to induce not only positive but also negative effects on metabolic conditions. For example, indoles produced from tryptophan induce interleukin-22 production, and, trimethylamine produced from choline is further converted into trimethylamine-oxide, which acts pro-inflammatory, while secondary bile acids converted from primary bile acids act anti-inflammatory, and SCFAs produced from fibers are involved in various metabolic pathways beneficially [13, 35-38]. However, although biological actions of these various metabolites have been evaluated extensively, nutrients consumption dependent changes in these metabolites have not well elucidated yet especially in relation with the gut microbiota. Namely, SCFAs have been reported to be increased in a diet-dependent fashion [13-16], but they are produced by not only proposed metabolically beneficial microorganisms but also by potentially pathogenic microorganisms [39, 40]. Therefore, nutrients consumption dependent association between DM and the gut microbiota observed here seem to warrant further analyses with more subjects and/or at the species or even subspecies levels.

Conclusion

At the end, please present the practical aspects of the study.

----Thanks. Upon your suggestion, we added the following sentence in conclusion:

   The results reported here seem to reinforce the importance of considering nutritional factors in examining the association between gut microbiota and DM and the joint effects of gut microbiota on that association, and may lead to discover which nutritional interventions are appropriate for which individuals in the future.

After implementing the above corrections please verify the abstract as well as make corrections to the relevant sections.

---We did.

Reviewer 3 Report

1. The authors use various expressions throughout the text that induce the idea of a cause-effect relationship (e.g., “influence of”, “risk for DM”, “protection against DM”, etc.). I advise a general rephrasing of the text, avoiding all expressions that suggest an “effect” phenomenon. Since the study’s cross-sectional nature can only inform us about associations and correlations, a cause-effect relationship can be hypothesised only but not completely supported and proven.

2. Details about the study population are insufficient. The original Iwaki study must be described extensively, and the authors should clarify the inclusion and exclusion criteria, aim and investigational pattern. At present, we do not know who the patients in this study are and what the study target and structure are.

3. The following phrase in the Introduction section is unclear and should be revised:

“Furthermore, because various intestinal microorganisms together seem to affect glucose metabolism jointly, analyses of the independent association of each taxon with DM may not be always reliable.”

Author Response

Dear Reviewer 3

Reviewer 3

   Thank you very much for having reviewed our manuscript entitled "Influence of nutrients consumed on associations between the gut microbiota composition and type 2 diabetes in a Japanese population:jcm-1717551”. We have changed our manuscript to fulfill your criticisms as much as possible, and in most cases, I just changed as you requested. But, in some cases where another reviewer recommended another way, you may see several changes that you have not suggested or no change that you have suggested. In any case, we beg your generosity to kindly feel satisfied with my responses. The details of the responses to each criticism were written below.

  1. The authors use various expressions throughout the text that induce the idea of a cause-effect relationship (e.g., “influence of”, “risk for DM”, “protection against DM”, etc.). I advise a general rephrasing of the text, avoiding all expressions that suggest an “effect” phenomenon. Since the study’s cross-sectional nature can only inform us about associations and correlations, a cause-effect relationship can be hypothesised only but not completely supported and proven.

------Thanks for your advice. Surely, your comment is appropriate, and, thus, we rephrased such expression as much as possible, especially in the sentences which describe results. Namely, we changed the words “risk for DM” and “protection against DM” in result parts throughout the text, although such words were left in some sentences with speculative comments, since we believe that without such explanation the points we would like to make most may become hard to be understood. In addition, for the words “influence of”, we could not find any suitable words to replace the words for the phrase “Influence of composition of selected nutrients consumed on associations between the gut microbiota composition and type 2 diabetes”. The title can be “Association between the gut microbiota composition and type 2 diabetes and composition of selected nutrients consumed “. However, we are afraid that such title does not seem to be the same written originally. So please forgive us for not rephrasing the words, “influence of”. Or, if possible, please tell me what to change the words in the sentence, so that we will change it as you advised.

  1. Details about the study population are insufficient. The original Iwaki study must be described extensively, and the authors should clarify the inclusion and exclusion criteria, aim and investigational pattern. At present, we do not know who the patients in this study are and what the study target and structure are.

-------Thanks. The original Iwaki study is just one of health care examination held in each community. No inclusion and exclusion criteria for participate are set. All residents over 20 years living in the are can participate if they want. To explain the facts, we added the following sentences to explain the Iwaki study more in details. In addition, we added a flow chart of the selection of study participants together with in/excluding criteria as Figure 1.  

"The Iwaki study is a population-based study with an annual comprehensive health check-up. Participants were recruited from the residents aged over 20 years living in the Iwaki area (number: about 9,200) through a public announcement. No inclusion and exclusion criteria are set for the Iwaki study per se."

  1. The following phrase in the Introduction section is unclear and should be revised:

“Furthermore, because various intestinal microorganisms together seem to affect glucose metabolism jointly, analyses of the independent association of each taxon with DM may not be always reliable.”

----Thanks for your comment. We revised it as follows to make what we would like to explain in the sentence clearer.

  "Furthermore, various intestinal microorganisms may not affect glucose metabolism solely but may affect it jointly, and, thus, analyses of the independent association of each taxon with DM may not be always reliable."

Reviewer 4 Report

Well presented the data from the authors, if it is possibile the farmacological treatment for type 2 diabetes, if there was any correlation with the presence of gut microbiota

Author Response

Dear Reviewer 4

   Thank you very much for having reviewed our manuscript entitled "Influence of nutrients consumed on associations between the gut microbiota composition and type 2 diabetes in a Japanese population: jcm-1717551”. We appreciate your evaluation a lot. The details of the responses to your comment were written below. Besides, you may see several changes that you have not suggested, which were made to reply to the comments raised by other reviewers. In any case, we beg your generosity to kindly feel satisfied with my responses.

Well presented the data from the authors, if it is possibile the farmacological treatment for type 2 diabetes, if there was any correlation with the presence of gut microbiota

----Thanks, we appreciate your evaluation a lot.

Yes, surely, your comment is very valuable. We would like to address to your above-mentioned comment if we can. However, we do not have any information of drugs taken for each subject, and, thus, we cannot examine such correlation. Therefore, we have described such fact as a limitations in discussion as follows: Drugs used by the study participants were also not evaluated, even though some drugs used to treat DM, such as metformin, …….. the influence of drugs by the study participants on the results reported here may not be so substantial.”

In addition, since the number of diabetic subjects is relatively small (n=96), we believe that such analyses may not override the problem of low statistic power. Therefore, such correlation is remained to be elucidated in the future.

So, please, be satisfied with the above-described explanation.

Round 2

Reviewer 3 Report

1. The new title lacks concision and clarity. Perhaps a more suitable version would be a formulation such as “Interrelations between gut microbiota composition, nutrient intake and diabetes status in an adult Japanese population”.

2. After changes in the text, the authors should re-examine the whole manuscript and rephrase sentences where the context is unclear. It is sometimes hard to understand at first reading if a specific section of the Results refers only to diabetic subjects or if the authors performed a pooled analysis on both the diabetic and control populations. Some isolated syntax errors are also present. Some of the phrases newly modified or introduced in the text are particularly prone to unclarity or errors; still, a thorough examination of the whole text would add value to the manuscript.

Author Response

Dear reviewer 3

   Thank you very much for having re-reviewed our manuscript entitled "Influence of nutrients consumed on associations between the gut microbiota composition and type 2 diabetes in a Japanese population: jcm-1717551”. We have changed our manuscript to fulfill your criticisms as much as possible, and in most cases, I just changed as you requested. So, we beg your generosity to kindly feel satisfied with my responses. The details of the responses to each criticism were written below.

  1. The new title lacks concision and clarity. Perhaps a more suitable version would be a formulation such as “Interrelations between gut microbiota composition, nutrient intake and diabetes status in an adult Japanese population”.

------Thanks for your comment. We appreciate your suggestion, and, thus changed the title as you suggested as follows:

Interrelations between gut microbiota composition, nutrient intake and diabetes status in an adult Japanese population

  1. After changes in the text, the authors should re-examine the whole manuscript and rephrase sentences where the context is unclear. It is sometimes hard to understand at first reading if a specific section of the Results refers only to diabetic subjects or if the authors performed a pooled analysis on both the diabetic and control populations. Some isolated syntax errors are also present. Some of the phrases newly modified or introduced in the text are particularly prone to unclarity or errors; still, a thorough examination of the whole text would add value to the manuscript.

-------Thanks. Since we did not have time to have English editing service for the phrases newly modified or introduced in the text, and, thus, such phrases might be made unclearly. We checked the whole manuscript and rephrase sentences where we think the context is unclear.  

  Besides, this study is a population-based cross-sectional study and thus, no results refer only to diabetic subjects. Although we had described such facts in methods, we believe that your comment is valuable, and, thus, but to make such facts clearer we added the following words in the results:

   The correlations between selected nutrients consumed by the subjects and their gut microbiota genera were then examined (Table 3).

Sincerely